# How well do deep neural networks trained on object recognition characterize the mouse visual system?

**Santiago A. Cadena,**[1,2,5,*] **Fabian H. Sinz,**[5,6] **Taliah Muhammad,**[3]
**Emmanouil Froudarakis,** [3,4] **Erick Cobos,** [3,4] **Edgar Y. Walker,** [5,6] **Jake Reimer,** [3,4]
**Matthias Bethge,** [1,2,5,†] **Andreas S. Tolias,** [3,4,†] **Alexander S. Ecker** [1,2,5,†,‡]

[1] Centre for Integrative Neuroscience, University of Tübingen, Germany
[2] Institute for Theoretical Physics, University of Tübingen, Germany
[3] Department of Neuroscience, Baylor College of Medicine, Houston, TX, USA
[4] Center for Neuroscience and Artificial Intelligence, Baylor College of Medicine, Houston, USA
[5] Bernstein Center for Computational Neuroscience, University of Tübingen, Germany
[6] Institute Bioinformatics and Medical Informatics (IBMI), University of Tübingen, Germany
† *Authors contributed equally*
‡ *Present address: Department of Computer Science, University of Göttingen, Germany*

* santiago.cadena@uni-tuebingen.de

## Abstract

Recent work on modeling neural responses in the primate visual system has benefited from deep neural networks trained on large-scale object recognition, and found a hierarchical correspondence between layers of the artificial neural network and brain areas along the ventral visual stream. However, we neither know whether such task-optimized networks enable equally good models of the *rodent* visual system, nor if a similar hierarchical correspondence exists. Here, we address these questions in the mouse visual system by extracting features at several layers of a convolutional neural network (CNN) trained on ImageNet to predict the responses of thousands of neurons in four visual areas (V1, LM, AL, RL) to natural images. We found that the CNN features outperform classical subunit energy models, but found no evidence for an order of the areas we recorded via a correspondence to the hierarchy of CNN layers. Moreover, the same CNN but with random weights provided an equivalently useful feature space for predicting neural responses. Our results suggest that object recognition as a high-level task does not provide more discriminative features to characterize the mouse visual system than a random network. Unlike in the primate, training on ethologically relevant visually guided behaviors – beyond static object recognition – may be needed to unveil the functional organization of the mouse visual cortex.

## 1 Introduction

Visual object recognition is a fundamental and difficult task performed by the primate brain via a hierarchy of visual areas (the ventral stream) that progressively untangles object identity information, gaining invariance to a wide range of object-preserving visual transformations [1, 2]. Fueled by the advances of deep learning, recent work on modeling neural responses in sensory brain areas builds upon hierarchical convolutional neural networks (CNNs) trained to solve complex tasks like object recognition [3]. Interestingly, these models have not only achieved unprecedented performance

33rd Conference on Neural Information Processing Systems (NeurIPS 2019), Vancouver, Canada. Neuro↔AI Workshop

in predicting neural responses in several brain areas of macaques and humans [4–7], but they also revealed a hierarchical correspondence between the layers of the CNNs and areas of the ventral stream [4, 6]: the higher the area in the ventral stream, the higher the CNN layer that explained it best. The same approach also provided a quantitative signature of a previously unclear hierarchical organization of A1 and A2 in the human auditory cortex [7].

These discoveries about the primate have sparked a still unresolved question: to what extent is visual object processing also hierarchically organized in the mouse visual cortex and how well can the mouse visual system be modeled using goal-driven deep neural networks trained on static object classification? This question is important since mice are increasingly used to study vision due to the plethora of available experimental techniques such as the ability to genetically identify and manipulate neural circuits that are not easily available in primates. Recent work suggests that rats are capable of complex visual discrimination tasks [8] and recordings from extrastriate areas show a gradual increase in the ability of neurons in higher visual areas to support discrimination of visual objects [9, 10].

Here, we set out to study how well the mouse visual system can be characterized by goal-driven deep neural networks. We extracted features from the hidden layers of a standard CNN (VGG16, [11]) trained on object categorization, to predict responses of thousands of neurons in four mouse visual areas (V1, LM, AL, RL) to static natural images. We found that VGG16 yields powerful features for predicting neural activity, outperforming a Gabor filter bank energy model in these four visual areas. However, VGG16 does not significantly outperform a feature space produced by a network with an identical architecture but random weights. In contrast to previous work in primates, our data provide no evidence so far for a hierarchical correspondence between the deep network layers and the visual areas we recorded.

## 2    Model Architecture

Our network (Fig.1) builds upon earlier work [5, 12]. It consist of four main network components: a *core* that provides nonlinear features of input images, a *readout* that maps those features to each neuron's responses, a *shifter* that predicts receptive field shifts from pupil position, and a *modulator* that provides a gain factor for each neuron based on running speed and pupil dilation of the mouse.

For the core we use VGG16 [11] up to one of the first eight convolutional layers. We chose VGG16 due to its simple feed-forward architecture, competitive object classification performance, and increasing popularity to characterize rodent visual areas [10, 13]. The collection of output feature maps of a VGG16 layer – the shared feature space – was then fed into a spatial transformer readout for each neuron (Fig.1B, see [12] for details). This readout learns one $(x, y)$ location for each neuron (its receptive field loca-

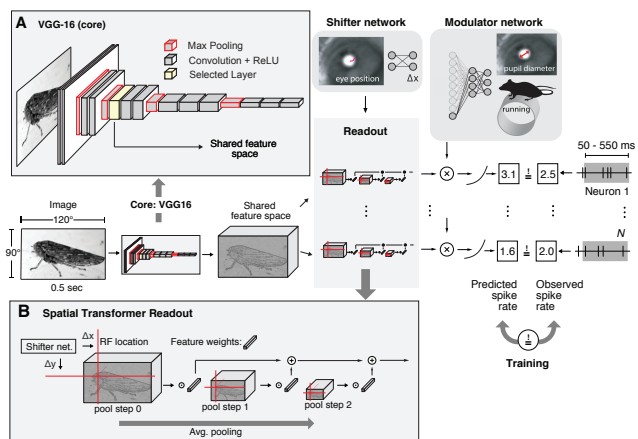

Figure 1:    VGG16 based model.  Input images are forwarded trough the *core* (**A**) network (first $n$ layers of VGG16) to produce a feature space shared by all neurons. Then, the spatial transformer *readout* (**B**) finds a mapping between these features and the neural responses for each neuron separately. The *shifter* network (an MLP with one hidden layer) corrects for eye movements. The output of the readout is multiplied by a gain predicted by the *modulator* network (an MLP with one hidden layer) that uses running speed and pupil dilation. A static nonlinearity converts the result into the predicted spike rate. All components of the model are trained jointly end-to-end to minimize the difference between predicted and observed neural responses.

tion, RF) [12] and extracts a feature vector at this location from multiple downsampled versions (scales) of the feature maps. The output of the readout is a linear combination of the concatenated feature vectors. We regularized the feature weights with an $L_1$ penalty to encourage sparsity.

Shifter and modulator are multi-layer perceptrons (MLP) with one hidden layer. The shifter takes the tracked pupil position in camera coordinates and predicts a global receptive field shift $(\Delta x, \Delta y)$

in monitor coordinates. The modulator uses the mouse's running speed, its pupil diameter, and the derivative to predict a gain for each neuron by which the neuron's predicted response is multiplied. A soft-thresholding nonlinearity turns the result into a non-negative spike rate prediction (Fig.1). All components of the model (excluding the core, which is pre-trained on ImageNet) are trained jointly end-to-end to minimize the difference between predicted and observed neural responses using Adam with a learning rate of $10^{-4}$, a batch size of 125 and early stopping.

## 3 Experiments

**Neural data.** We recorded responses of excitatory neurons in areas V1, LM, AL, and RL (layer 2/3) from two scans from one mouse and a third scan from a second mouse with a large-field-of-view two-photon mesoscope (see [14] for details) at a frame rate of 6.7 Hz. We selected cells based on a classifier for somata on the segmented cell masks and deconvolved their fluorescence traces, yielding 7393, 4674, 4680, 5797 neurons from areas V1, LM, AL, and RL, respectively. We further monitored pupil position, pupil dilation, and absolute running speed of the animal.

**Visual stimuli.** Stimuli consisted of 5100 images taken from ImageNet, cropped to 16:9 and converted to gray-scale. The screen was $55 \times 31$ cm at a distance of 15 cm, covering roughly $120° \times 90°$. In each scan, we showed 5000 of these images once (training and validation set) and the remaining 100 images 10 times each (test set). Each image was presented for 500 ms followed by a blank screen lasting between 300 ms and 500 ms. For each neuron, we extract the accumulated activity between 50 ms and 550 ms after stimulus onset using a Hamming window.

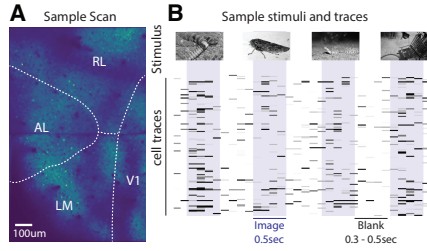

Figure 2: Neural data. **A**. Example large-field-of-view scan. **B**. Visual paradigm with sample cell traces

## 4 Results

We fitted one model (that of Fig.1; see [12] for training details) for each combination of scan, brain area, VGG16 layer (out of the first eight), random initialization (out of three seeds), and input resolution. We considered several resolutions of the input images because the right scale at which VGG16 layers extract relevant features that best match the representation in the brain is unknown. Optimizing the scale for each layer was critical, since the correspondence between a single layer and a brain area (in terms of best correlation performance) strongly depends on the input resolution (e.g. see Fig 3A for V1 data). For further analyses (Fig 3B & 4), we pick for each case the best performing input scale in the validation set.

**No hierarchical correspondence.** Previous results in primates [4] show that a brain area higher in the hierarchy is better matched (i.e has a peak in prediction performance) by a higher network layer. In contrast, when comparing the average performance across cells and scans for each convolutional layer and brain area, we find no clear evidence for a hierarchy (Fig.3B) since there is no clear ordering of the brain areas.

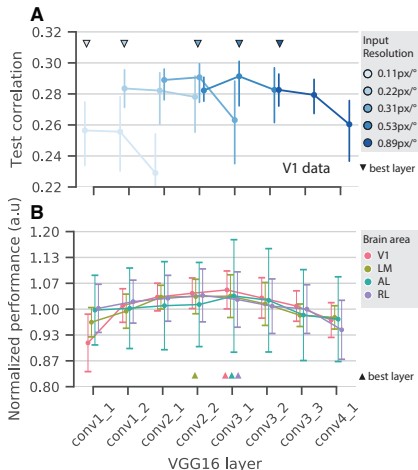

Figure 3: No clear hierarchical correspondence. **A** Test correlation in V1 data for different input resolutions as a function of VGG16 layer. **B** Normalized performance for all four brain areas. Triangles show the best predictive layer in each case

**VGG16 outperforms classical models.** We then investigate whether the lack of an evident hierarchy was due to an overall poor performance of our model. Thus, we first revise how much of the explainable stimulus-driven variability the VGG16-based model captures. To this end we calculate the oracle correlation (the conditional mean of $n-1$ responses without the model) [12] obtaining an upper bound of the achievable performance. Then we evaluate the test correlation of our model

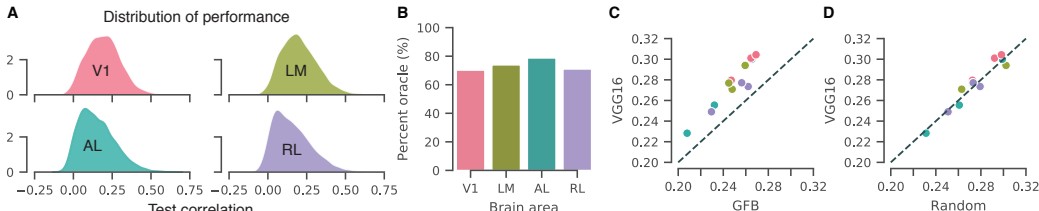

Figure 4: Performance comparison. **A**. Density of the distribution of test correlation across all neurons. **B**. Percentage of oracle performance. In **C** and **D**, each point is the average test correlation performance for one scan across all neurons. Brain areas are color coded as in **A** and the dotted line represents the identity. **B**: VGG16 vs. GFB. **C**: VGG16 vs. Random network core with the same architecture of VGG16

restricted to visual input information (no shifter and no modulator), against the oracle (Fig.4B) and find that VGG16 features explain a substantial fraction of the oracle for the for areas (70–78%)

Second, we consider a subunit energy model with Gabor quadrature pairs as a baseline due to its competitive predictive performance of macaque V1 responses [5]. We replace the core from Fig. 1 with a Gabor filter bank (GFB) consisting of a large number of Gabor filters with different orientations, sizes and spatial frequencies arranged in quadrature pairs, and followed by a squaring nonlinearity [5]. We find that for all areas and scans, the VGG16 core outperformed the GFB (Fig.4C).

**Core with random weights performs similarly.** The results so far show that VGG16 provides a powerful feature space to predict responses, which may suggest that static object recognition could be a useful high-level goal to describe the function of the mouse visual system. However, we were surprised that most VGG layers led to similar performance. To understand this result better, we also evaluated a core with identical architecture but random weights. This random core performed similarly well as its pre-trained counterpart (Fig.4D), suggesting that training on static object recognition as a high-level goal is not necessary to achieve state-of-the-art performance in predicting neural responses in those four visual areas. Instead, a sufficiently large collection of random features followed by rectification provides a similarly powerful feature space.

**The number of LN layers is critical to best match neural activity.** Since random features produced by a linear-nonlinear (LN) hierarchy closely match the performance of the pretrained VGG16, we then asked if the number of LN steps – when accounting for multiple input resolutions – was the key common aspect of these networks that yielded the best predictions. Effectively, similar to the case of the pretrained VGG16 core, we found that the fourth and fifth rectified convolutional layers of the random core are the best predictive layers for the four areas we studied (Fig. 5). However, it is important to note that in both cases the increase in performance after the second convolutional layer is only marginal. Overall, we conclude that the nonlinear degree – number of LN stages – rather than the static object recognition training goal dictates how close the representations are to the neural activity.

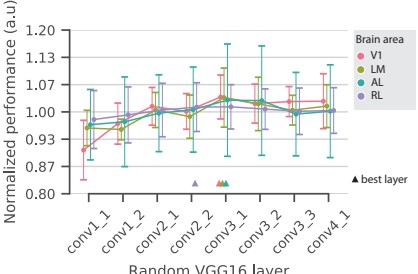

Figure 5: Normalized performance of the random core with VGG16 architecture for all four brain areas. Triangles show the best predictive layers in each case

## 5 Discussion

In contrast to similar work in the primate, we find no match between the hierarchy of mouse visual cortical areas and the layers of CNNs trained on object categorization. Although VGG16 achieves state-of-the-art performance, it is matched by random weights. There are three implications of our results: First, our work is in line with previous work in machine learning that shows the power of random features [15]. Therefore, we argue that models based on random features should always be reported as baselines in studies on neural system identification. Second, which VGG layer best predicted any given brain area depended strongly on the image resolution we used to feed into VGG16.

We observed a similar effect in our earlier work on primate V1 [5]. Thus, the studies reporting a hierarchical correspondence between goal-driven deep neural networks and the primate ventral stream should be taken with a grain of salt, as they – to the best of our knowledge – do not include this control. Third, optimizing the network for static object recognition alone as a high-level goal does not appear to be the right approximation to describe representations and the visual hierarchy in the mouse cortex. Although our results do not exclude a potential object processing hierarchy in the mouse visual system, they suggest that training with more ethologically relevant visually guided tasks for the mouse could be a more fruitful goal-driven approach to characterize the mouse visual system [16]. For instance, an approach with dynamic stimuli such as those found during prey capture tasks [17] could yield more meaningful features to unveil the functional organization of the mouse visual system.

**Acknowledgments**    S.A.C was supported by the International Max Planck Research School for Intelligent Systems (IMPRS-IS). The research was supported by the German Federal Ministry of Education and Research (BMBF) via the Competence Center for Machine Learning (FKZ 01IS18039A); the German Research Foundation (DFG) grant EC 479/1-1 (A.S.E.), the Collaborative Research Center (SFB 1233, Robust Vision) and the Cluster of Excellence "Machine Learning – New Perspectives for Science" (EXC 2064/1, project number 390727645); the Bernstein Center for Computational Neuroscience (FKZ 01GQ1002); the Intelligence Advanced Research Projects Activity (IARPA) via Department of Interior/Interior Business Center (DoI/IBC) contract number D16PC00003. F.S. is supported by the Institutional Strategy of the University of Tübingen (Deutsche Forschungsgemeinschaft, ZUK 63), the Carl-Zeiss-Stiftung, and Amazon AWS through a Machine Learning Research Award. The U.S. Government is authorized to reproduce and distribute reprints for Governmental purposes notwithstanding any copyright annotation thereon. Disclaimer: The views and conclusions contained herein are those of the authors and should not be interpreted as necessarily representing the official policies or endorsements, either expressed or implied, of IARPA, DoI/IBC, or the U.S. Government.

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
