# OpenReview forum: "How well do deep neural networks trained on object recognition characterize the mouse visual system?"
_NeurIPS.cc/2019/Workshop/Neuro_AI — Real Neurons & Hidden Units @ NeurIPS 2019 Oral_

### Official Review · AnonReviewer3 · 2019-09-24
**Necessary work competently executed**

**Clarity:** 4

**Category:**

AI->Neuro

**Clarity Comment:**

The presentation is mostly clear and straightforward. One point of confusion as to how the model was trained: the paper says "All components of the model are trained jointly end-to-end". I initially believe the VGG core wasn't trained, but this statement made that unclear.

**Evaluation:**

5: Excellent

**Importance:**

4: Very important

**Importance Comment:**

CNNs have been explored thoroughly with primate and human data. Given the extensive use of mice in neuroscience research it is very important to understanding how these models relate to those systems and ideally how to build a good model for mouse data.

**Intersection:**

4: High

**Intersection Comment:**

This work directly compares a task-trained ML model to mouse data and so is directly at the intersection of ML and neuro.

**Rigor Comment:**

The authors seem to take into account all the relevant complexities including image scale, RF size, running speed, etc. I would be interested to see how the results would differ if trying to train and predict on average responses over many trials of the same image, as this would be a more direct comparison to what is done in primate research.
The random weight test was an important and insightful control to do.

**Technical Rigor:**

5: Near certainty

---

> ### Author Response · Authors · 2019-10-30
> **Thank you for your positive review.**
>
> - Regarding training and testing on average responses: Note that we cannot train on averages, as we do not have repeats in the training set (also, most of the monkey work does not train on averages). We could do the evaluation on averages, but we (a) do not think it matters for our conclusions what evaluation metric is used, because we are comparing different models on the *same* data, (b) averages of 10 trials on noisy calcium imaging data will still contain noise unrelated to the stimulus and (c) the numbers are not comparable to monkey work anyway due to differences in recording technique (calcium imaging vs. electrophysiology), stimulus presentation time, brain areas, species etc.
>
> - Regarding your clarity comment, we changed that sentence to "All components of the model (excluding the core, which is pre-trained on ImageNet) are trained jointly end-to-end."

---

### Official Review · AnonReviewer2 · 2019-09-26
**Important question, well addressed, with an interesting result**

**Clarity:** 5

**Comment:**

-- Generally, great paper. Clear presentation of thorough work, exploring an important question.

-- Would have been great to include another Imagenet-trained architecture, since different architectures have widely varying macaque brain predictivity, and that of VGG16 is not particularly high (Schrimpf et al., 2018 BrainScore).

-- I'm not a big fan of the asterisks in Figures 3A and 3B used to indicate the best layers in various model tests. It doesn't provide any additional information to the data lines themselves, and it leads the reader to expect these indicate statistically significant comparisons.

-- Typo page 4 line 158: "pray" >> "prey"

**Category:**

AI->Neuro

**Clarity Comment:**

-- Very well written. Figures exceptionally detailed and thoroughly labelled. Methods described clearly and in good detail.

**Evaluation:**

5: Excellent

**Importance:**

4: Very important

**Importance Comment:**

-- The surprisingly high power of randomly weighted DCNNs is a point that has popped up a couple of times in recent human fMRI / MEG work. The present paper makes the important case that random networks should be included as a matter of course in DCNN modelling projects, and sounds a note of caution about the field's temptation to over-interpret the particular features learned by high-performing trained networks.

**Intersection:**

3: Medium

**Intersection Comment:**

-- Mostly neuroscientific, but addresses the important topic of how models from machine learning can best be used in neuro research.

**Rigor Comment:**

-- Comprehensive data measurement and modelling pipeline. Use of the same spatial transformer model with an interchangeable bank of input features is elegant.

**Technical Rigor:**

5: Near certainty

---

> ### Author Response · Authors · 2019-10-30
> **Thank you for positive and constructive review**
>
> We picked VGG16 for practical purposes, as it is a simple architecture and has been used in other studies of the mouse visual system [refs]. We agree that we would have to test additional models to fully establish the negative result that goal-driven training on object recognition is not sufficient to explain the mouse visual system.
>
> However, we think such an extensive analysis is beyond the scope of this workshop paper: to be able to make solid statements about the usefulness of a model is important to carefully optimize the readout for each model, which requires a substantial amount of resources in terms of both engineering and computation time.
>
> Thanks for pointing out that the asterisks in Fig. 3 might confuse the reader. We replaced them by triangles, but opted to keep them, because they highlight the specific points panels A and B are making.

---

### Official Review · AnonReviewer1 · 2019-09-26
**well-constructed models and comparisons to large-scale neural recordings in mouse V1**

**Clarity:** 5

**Comment:**

Would have liked to know more about which layers of the random network corresponded to neural activity, as a sort of benchmark of how many non-linear RELUs the pixels might need to go through before obtaining a representation that's as similar to neural activity as VGG.

**Category:**

AI->Neuro

**Clarity Comment:**

Very clear.

**Evaluation:**

5: Excellent

**Importance:**

5: Astounding importance

**Importance Comment:**

Object recognition networks have been the benchmark model for a long time, but other options have not been explored in depth. This paper points out that networks with random weights fit mouse visual activity just as well as models trained to perform object recognition tasks, suggesting that the object recognition task itself is probably not an ethological comparison.

**Intersection:**

4: High

**Intersection Comment:**

Strong intersection.

**Rigor Comment:**

Thoroughly thought-out model and fitting procedure.

**Technical Rigor:**

5: Near certainty

---

> ### Author Response · Authors · 2019-10-30
> **Thanks for the positive review**
>
> Thank you for reviewing our paper and highlighting its important implications. We added the information about how different layers of the random network perform in the final version (Figure 5 and corresponding paragraph). Overall, the number of layers that best match the neural data in a random network was the same as with the trained network. This result suggests that for this dataset, the number of LN stages is what's most critical to predict neural responses (given the number of features in each layer).

---

### Decision · Program_Chairs · 2019-10-02

Accept (Oral)